

# The effect of ethanol concentration on the morphological and molecular preservation of insects for biodiversity studies

Daniel Marquina[1,2], Mateusz Buczek[3], Fredrik Ronquist[1] and Piotr Łukasik[1,3]

[1] Department of Bioinformatics and Genetics, Swedish Museum of Natural History, Stockholm, Sweden
[2] Department of Zoology, Stockholm University, Stockholm, Sweden
[3] Institute of Environmental Sciences, Faculty of Biology, Jagiellonian University, Krakow, Poland

Corresponding author
Daniel Marquina,
daniel.marquina@nrm.se

## ABSTRACT

Traditionally, insects collected for scientific purposes have been dried and pinned, or preserved in 70% ethanol. Both methods preserve taxonomically informative exoskeletal structures well but are suboptimal for preserving DNA for molecular biology. Highly concentrated ethanol (95–100%), preferred as a DNA preservative, has generally been assumed to make specimens brittle and prone to breaking. However, systematic studies on the correlation between ethanol concentration and specimen preservation are lacking. Here, we tested how preservative ethanol concentration in combination with different sample handling regimes affect the integrity of seven insect species representing four orders, and differing substantially in the level of sclerotization. After preservation and treatments (various levels of disturbance), we counted the number of appendages (legs, wings, antennae, or heads) that each specimen had lost. Additionally, we assessed the preservation of DNA after long-term storage by comparing the ratio of PCR amplicon copy numbers to an added artificial standard. We found that high ethanol concentrations indeed induce brittleness in insects. However, the magnitude and nature of the effect varied strikingly among species. In general, ethanol concentrations at or above 90% made the insects more brittle, but for species with robust, thicker exoskeletons, this did not translate to an increased loss of appendages. Neither freezing the samples nor drying the insects after immersion in ethanol had a negative effect on the retention of appendages. However, the morphology of the insects was severely damaged if they were allowed to dry. We also found that DNA preserves less well at lower ethanol concentrations when stored at room temperature for an extended period. However, the magnitude of the effect varies among species; the concentrations at which the number of COI amplicon copies relative to the standard was significantly decreased compared to 95% ethanol ranged from 90% to as low as 50%. While higher ethanol concentrations positively affect long-term DNA preservation, there is a clear trade-off between preserving insects for morphological examination and genetic analysis. The optimal ethanol concentration for the latter is detrimental for the former, and vice versa. These trade-offs need to be considered in large insect biodiversity surveys and other projects aiming to combine molecular work with traditional morphology-based characterization of collected specimens.

## INTRODUCTION

The first records of the use of ethanol for the preservation of animal tissue date back to the mid 1600s, when Robert Boyle mentions that he successfully used the ''Spirit of Wine'' to preserve blood and soft parts of a human body, as well as a fish, for many months (*Boyle, 1664*). The knowledge of the preservative properties of ethanol, in combination with the discovery of cheap and effective ways of producing ethanol in high concentration, led to widespread adoption among naturalists. Consequently, ethanol has been the fixative most widely used in museum and private natural history collections since the 18th century. The attractiveness of ethanol for other human uses has sometimes caused challenges, however. For instance, Carl Linnæus, the father of modern taxonomy, needed to apply for a special permission to import ethanol for preserving his collections after home distilling was temporarily banned in Sweden (*Von Linné, 1764*).

During the last two decades, as the research focus has shifted from the analysis of morphological features to molecular work, including DNA sequencing and amplification, ethanol has remained the preferred preservative liquid. Ethanol is an excellent fixative for DNA for three reasons: it kills decomposing microorganisms; it removes water from the tissues, slowing down enzymatic processes; and it denatures both the DNA and DNA-degrading enzymes, preventing further enzymatic degradation (*Srinivasan, Sedmak & Jewell, 2002*). For insects, other preservative liquids, for example acetone (*Fukatsu, 1999*), or fundamentally different approaches such as silica-drying or ultrafreezing (<-70 °C), preserve the DNA as well as ethanol (*Carvalho & Vieira, 2000*; *Post, Flook, & Millest, 1993*). These methods, nonetheless, are not always a viable alternative, especially during long and remote collection campaigns, and ethanol does remain the standard preservative in biodiversity research.

The typical ethanol concentration employed for preserving insects for morphological examination used to be 70% (*Martin, 1977*). However, a higher concentration (95% or higher) has been recommended for the optimal preservation of DNA (*Nagy, 2010*). Although 70% or 80% ethanol is known to suffice as a DNA preservative for PCR and sequencing purposes during short-term storage (*Carew, Coleman & Hoffmann, 2018*; *Stein et al., 2013*), it has been shown that DNA degradation occurs in the longer term, resulting in the molecule getting increasingly fragmented over time (*Baird et al., 2011*; *Carew et al., 2017*). Thus, it would appear advantageous to change the standard practice and preserve insects in higher ethanol concentrations. This idea has not been universally adopted, however, because of the common assumption that high grade ethanol makes the insects brittle and prone to damage during manipulation, most likely due to excessive tissue dehydration. However, to our knowledge, there is only one published study that has directly addressed the effect of high ethanol concentrations on insect preservation for morphological study: *King & Porter (2004)* used three species of ants to test the effects of ethanol concentration on the mounting of specimens, concluding that 95% ethanol made the ants hard to mount and prone to breaking. Our own experience with other insect

groups is similar. For instance, we noted that cicadas (Hemiptera) become very brittle when preserved in 95% ethanol (repeatedly replaced to prevent its dilution by insect body fluids) and subsequently mounted. Preserving them in 90% ethanol made cicadas much more  suitable for mounting and taxonomic characterization (K Nazario and P Łukasik, 2016, pers. obs).

Increasingly, insect biodiversity inventories and ecological assessments are accompanied by or solely rely on DNA sequencing (e.g., *Janzen et al., 2009*; *Shokralla et al., 2014*; see *Matos-Maraví et al. (2019)*) for a comprehensive review on insect genomics applied to biodiversity study). This leads researchers to collect samples in the field in higher concentrations of ethanol, which could be detrimental for the morphological integrity of the insects collected, potentially causing the loss of morphological characters and decreasing the value of the specimens for taxonomic description or further anatomical or ecological study. When the purpose is to store specimens for both DNA sequencing and morphological study, researchers seem to be using a variety of approaches: from standardizing ethanol concentrations to 80%, through opting for concentrations closer to 95%, to starting with a high concentration and not measuring it again after insects are preserved, or relying on sub-zero temperatures as the primary means of preservation. There appears to be no hard evidence or consensus on which approach is optimal.

While it may be challenging to find a sweet spot that is acceptable for all intended uses, there is a clear need for experimental studies of the nature of the trade-offs between the preservation of insects for DNA sequencing and morphological study. Here, we take a few steps towards filling this knowledge gap. Specifically, we examined the effects of increasing ethanol concentrations on specimen fragility and DNA preservation in seven species of insects, spanning four orders and representing different sizes and levels of sclerotization.

## MATERIAL AND METHODS

### Mock communities

For the experiments, we constructed artificial (mock) communities made up of seven species of insects (Fig. 1A): *Macrolophus pygmaeus* (Hemiptera: Miridae), *Aphidoletes aphidimyza* (Diptera: Cecidomyiidae), *Drosophila hydei* (Diptera: Drosophilidae), *Dacnusa sibirica* (Hymenoptera: Braconidae), *Calliphora vomitoria* (Diptera: Calliphoridae), *Formica rufa* (Hymenoptera: Formicidae) and *Dermestes haemorrhoidalis* (Coleoptera: Dermestidae). For better readability, only the generic names will be used from now on in the article. Adults of these species represent a wide range of body shapes, cuticle hardness levels, and responses to varying ethanol concentrations and treatments based on anecdotal information and prior observations. Specimens of some species were commercially purchased, and others were manually collected (Table S1). Specimens of *Macrolophus*, *Drosophila*, *Dacnusa*, *Calliphora*, and *Formica* were first killed by freezing them at −20 °C for 1–2 h and then placed in the experimental tubes at the desired ethanol concentration (see below). Specimens of *Aphidoletes* and *Dermestes* were killed by placing them in the experimental tubes at the desired concentrations directly. All mock communities consisted of ten individuals of *Macrolophus*, *Aphidoletes*, *Drosophila* and *Dacnusa*, and two individuals of *Calliphora*,

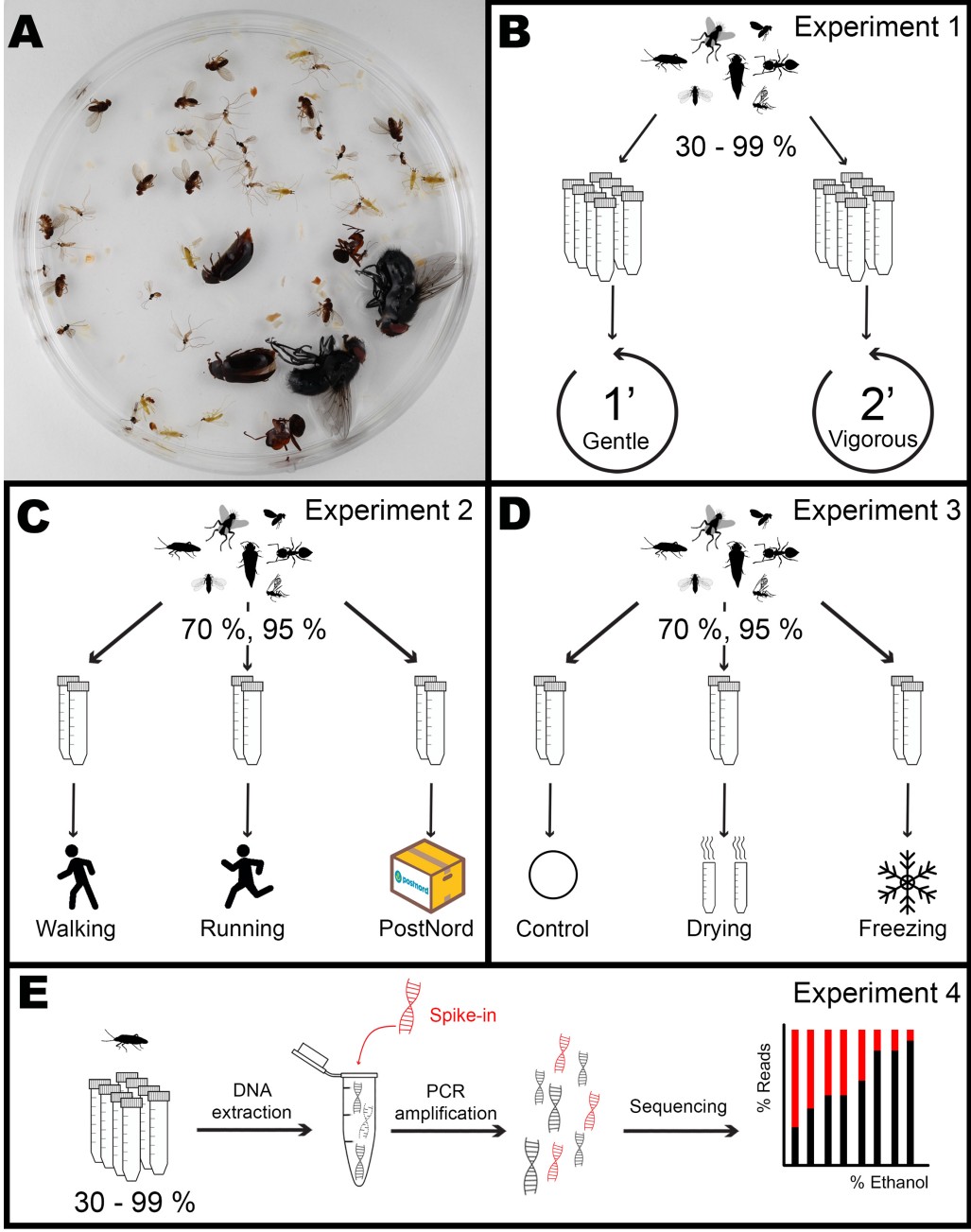

**Figure 1 Overview of the experimental design.** Each mock community sample (A) was assigned to an ethanol concentration and experiment. In Experiment 1 (B), five tubes for each of eight ethanol concentrations (30, 50,70, 80, 90, 95, 97, and 99%) were subjected to Gentle or Vigorous shaking. Inn Experiment 2 (C), three tubes for each of two ethanol concentrations (70, 95%) were either carried by a walking or a running experimenter or sent by the Swedish national post service (PostNord) in two different parcels. In Experiment 3 (D), four tubes for each of two ethanol concentrations (70, 95%) were shaken under a Gentle regime after being either dried, (continued on next page…)

**Figure 1 (…continued)**
frozen or left as control. After each treatment, the number of appendages lost by every individual was
scored. In Experiment 4 (E), DNA was extracted from one individual of each species from every tube in
treatment Gentle from Experiment 1, mixed with a known quantity of synthetic DNA, PCR amplified
and sequenced, and the proportion between insect DNA and synthetic DNA calculated as a measure of
DNA degradation. Image credits: Dave Angelini (http://phylopic.org/image/ed1b3f0f-2a19-4fc9-8525-
bbfe50e28d1c/), Creative Commons Attribution 3.0 Unported license; T. Michael Keesey (http://phylopic.
org/image/3010e744-6b06-4022-9ce6-354db93e99fe/), Public Domain Dedication 1.0 license; Ramiro
Morales-Hojas (http://phylopic.org/image/48bcbd0d-a902-4be8-9acc-60a6fbfd5f8d/), Public Domain
Dedication 1.0 license; Maxime Dahirel (digitisation), Kees van Achterberg et al. (doi: 10.3897/BDJ.8.
e49017) (original publication) (http://phylopic.org/image/a7fd469a-3835-4ccf-a663-71a8b7817916/),
Creative Commons Attribution 3.0 Unported license; collnell (http://phylopic.org/image/7252c46a-6bf1-
42dd-ac5a-1018f404dfc8/, http://phylopic.org/image/7252c46a-6bf1-42dd-ac5a-1018f404dfc8/), Pub-
lic Domain Dedication 1.0 license; T Michael Keesey (after Ponomarenko) (http://phylopic.org/image/
27356f15-3cf8-47e8-ab41-71c6260b2724/), Public Domain Dedication 1.0 license.

*Formica,* and *Dermestes*. The specimens were kept in 50 mL Falcon tubes with 40 mL of
preservative ethanol of different concentrations. Communities were prepared over the
course of approx. 2 weeks. Once all communities were ready, the ethanol was replaced
with a fresh aliquot of ethanol at the same concentration and kept for a month at room
temperature to standardize the incubation before the treatments began.

We used these preserved mock communities for three experiments, where we measured
the effect of ethanol concentration on the morphological integrity of specimens subjected to
different handling regimes, as described below. Insects used for the first of these experiments
were subsequently stored long-term at the same ethanol concentration and used for the
DNA preservation study.

## Experiment 1: effect of ethanol concentration on specimen brittleness

In the first experiment, we analyzed whether short-term preservation in high concentrations
of ethanol alone increased the fragility of the insects. For the main experiment (Fig. 1B),
we used mock communities preserved in eight ethanol concentrations: 30, 50, 70, 80, 90,
95, 97, and 99%, with ten replicate tubes for each concentration. The tubes were subjected
to two different shaking regimes by manually vortexing them in a horizontal position
for either 1 min ("Gentle" shaking, five tubes per concentration) or 2 min ("Vigorous"
shaking, the remaining five tubes per concentration). After each treatment, all individuals
were inspected under a stereomicroscope and scored for lost appendages. The type of
appendages scored for each species varied, and this was decided based on pilot trials (see
Supplementary Material). Specifically, we scored the loss of: head, legs, wings, and antennae
for *Macrolophus*, *Aphidoletes,* and *Dacnusa*; head, legs, and wings for *Drosophila*; legs and
wings for *Calliphora* and *Dermestes*; and legs and antennae for *Formica*. Only forewings
were considered (elytra in the case of *Dermestes*).

## Experiment 2: effect of transport regimes

In the second experiment, we measured if the sensitivity to transport-induced damage
was influenced by the ethanol concentration. For this experiment (Fig. 1C), we used two
ethanol concentrations: 70 and 95%, with 12 mock community replicates in each. These
tubes were transported in three ways, corresponding to treatments. Treatment "Walking"

consisted of the experimenter (D.M.) carrying three tubes/concentration in the backpack home-to-work and work-to-home for a total distance of approx. 10 km (three days * 3.2 km), walking only. Treatment "Running" was similar, but the experimenter (P.Ł.) ran, chasing after public transportation, for approximately half of the total distance of approx. 8 km (three days * 2.7 km). The last treatment, "PostNord", consisted of two shipments of 3 tubes/concentration, separated by one week, from the Swedish Museum of Natural History in Stockholm to Station Linné in Öland and back, using the Swedish national post service. We reasoned that these three treatments would represent the range of handling regimes that insect samples collected in the field might experience in practice. Damage induced to the individual insects was assessed as above.

## Experiment 3: effect of storage and processing factors

In the third experiment, we tested if other types of treatment can magnify the effects of high ethanol concentrations on insect brittleness (Fig. 1D). For each of the two ethanol concentrations tested (70 and 95%), we used 12 replicate mock communities, four tubes/concentrations, for each of the three pre-treatments. The first pre-treatment, "Freezing", was chosen to represent the effects of repeated freeze-thaw cycles that some Malaise trap samples may be experiencing during long-term storage. Specifically, we subjected our samples to three cycles of 16 h at −20 °C and 8 h at room temperature. The second pre-treatment, "Drying", tests the effects of specimen drying, included in some non-destructive DNA extraction protocols (e.g., *Nielsen et al., 2019*; *Vesterinen et al., 2016*) prior to digestion in the lysis buffer. Samples may also dry up accidentally, especially in difficult field conditions. In this treatment, we carefully poured away the ethanol from the tube, dried the insects for 24 h in the tube at room temperature, and afterwards, added 40 mL of ethanol at the original concentration. The last treatment was a control: the communities remained in ethanol at room temperature. After the pre-treatment, all tubes were manually vortexed in a horizontal position for 1 min, corresponding to the "Gentle" treatment described in Experiment 1. The damage was assessed and scored as in Experiments 1 and 2.

## Experiment 4: effect of ethanol concentration on DNA preservation

In the last experiment, we used insect samples from the first experiment to test the effects of ethanol concentration on DNA preservation (Fig. 1E). After the brittleness assessment, we placed the insects back in the tubes with their corresponding ethanol concentration and stored them at room temperature, in shaded boxes, for twelve months. Then, for each of the seven experimental species preserved in eight ethanol concentrations (30, 50, 70, 80, 90, 95, 97, and 99%), we took a single insect specimen (*Dacnusa*) or a single insect leg (other species) from each of the five tubes from the "Gentle" treatment. We reasoned that in specimens stored at lower ethanol concentrations after extended storage, DNA is likely to be more degraded, containing fewer amplifiable copies of target genes in the same amount of insect tissue. To test this, we quantified differences in PCR template amounts across ethanol concentrations by adding the same amount of an artificial target to each sample of a given species prior to DNA extraction. Subsequently, we used the extracted
DNA to prepare cytochrome oxidase I (COI) amplicon libraries that were then sequenced and assessed the ratios of an artificial target to insect COI gene in the resulting amplicon datasets.

## Amplicon library preparation and bioinformatic processing (Experiment 4)

We placed each sample (one entire specimen for *Dacnusa* or a single leg for the rest of species) in a two mL tube, containing 190 µL lysis buffer (0.4M NaCl, 10 mM Tris–HCl, 2 mM EDTA, 2% SDS), 10 µL of proteinase K and ceramic beads (2.8 mm and 0.5 mm). We homogenized the samples on an Omni Bead Ruptor 24 homogenizer during two 30 s cycles at the speed of 5 m/s and then we incubated them at 56° C for 2 h. From this homogenate, we mixed 40 µL (corresponding to 20% of the total lysate volume) with a predefined amount of a linearized spike-in plasmid carrying an artificial COI target. The plasmid contained a 471-base insert developed based on the *Calliphora vomitoria* barcode region sequence, where outside of the conserved primer regions, approximately 40% of bases were substituted. We determined plasmid quantities were determined for each species during pilot experiments and ranged from 11,000 to 317,000 copies per sample (Table S7). We purified DNA by mixing 42 µL of each sample (40 µL of homogenate and 2 µL of plasmid) with 80 µL of home-made Sera-mag SpeedBead solution, separating the beads on a magnetic plate, washing with 80% ethanol twice, and finally, resuspending the DNA in 20 µL of TE buffer.

We prepared COI amplicon libraries for 278 samples, plus six negative controls (two for the lysis buffer, two for PCR, and two for indexing), following a two-step PCR library preparation protocol (Method 4, (*Glenn et al., 2019*). In the first PCR, the 458-bp target region was amplified using template-specific primers with variable-length inserts and Illumina adapter tails. After the bead-purification of products, products were completed with Illumina adapters unique for each library in the second, indexing PCR. The oligonucleotide sequences and reaction conditions are provided in Supplementary Material S5. The pooled libraries were sequenced in a multiplexed Illumina MiSeq v3 lane (2 ×300 bp reads) at the Institute of Environmental Sciences of Jagiellonian University (Kraków, Poland).

We processed the amplicon data using Mothur v1.44.2 (*Schloss et al., 2009*). Reads were merged into contigs, had primers trimmed, were dereplicated, and singleton genotypes were removed. These filtered data were then used for 97% OTU picking. The commands used are provided in the Supplementary Material. Next we identified the dominant OTUs corresponding to target insects and the artificial target, and for all samples, calculated the insect OTU to artificial target OTU ratio. Additionally, because for three species (*Drosophila*, *Formica*, and *Macrolophus*) additional COI OTUs (likely corresponding to nuclear mitochondrial pseudogenes) were relatively abundant, we repeated the analyses after summing up the reads of all OTUs that were at least 90% identical to the dominant sequence. We then translated the abundance ratios to an estimated number of insect COI copies in each sample and standardized the values relative to the median value for 95% ethanol concentration for that same species.

## Statistical analyses

We conducted all analyses in R (*R Development Core Team, 2017*) using the base package and the packages 'glmmTMB' (*Brooks et al., 2017*), 'DHARMa' (*Hartig, 2019*), and 'emmeans' (*Lenth, 2018*). Visualizations were generated with package 'ggplot2' (*Wickham, 2016*). For the morphological preservation analysis, we considered all types of appendages together (Appendages = Legs + Wings + Antennae + Head), and analyzed each species separately for each experiment. If the head was lost, we scored the antennae also as lost. A generalized linear mixed effects model was fitted to the data with the number of appendages lost as a function of Treatment and Concentration, and their interaction, as fixed effects, and Tube (= replicate) as a random effect, assuming a log-link and Poisson distribution (function *glmmTMB* from package glmmTMB). As some species did not lose any appendages at all in certain treatments, zero inflation in the model was tested by simulating scaled residuals with the function *simulateResiduals* (package DHARMa) with the model without considering zero inflation, and checking the presence of zero inflation (function *testZeroInflation* from package DHARMa). If significant, a term for controlling zero inflation dependent on Concentration was included in the model. Subsequently, we used an analysis of variance (ANOVA) to determine whether Treatment, Concentration, and their interaction had a significant effect on the number of lost appendages (function *Anova.glmmTMB* from package glmmTMB), and we calculated Tukey-adjusted pairwise differences between treatments (function *emmeans* from package emmeans).

For the DNA preservation analysis, we considered a linear model was fitted to the data with a logarithm-transformed estimated number of COI target copies (relative to 95% ethanol concentration median) as a function of two factors: Species and Ethanol Concentration, and their interaction. As both factors and their interaction were statistically significant (Species: $df = 6$, $F = 88.31$, $p < 0.001$; Ethanol concentration: $df = 7$, $F = 41.66$, $p < 0.001$; Species x Ethanol Concentration: $df = 42$, $F = 3.81$, $p < 0.001$), we analyzed data from each species separately. In cases where the estimated number of COI target copies depended significantly on ethanol concentration, contrasts were used to test for differences between each ethanol concentration and 95% ethanol concentration treatment, assumed as the reference. We conducted the same analyses on the second dataset, containing all OTUs at >90% identity to the reference sequence.

# RESULTS

## Experiment 1: Effects of ethanol concentration on specimen brittleness

For most insect species tested, we found a significant effect of ethanol concentration on the number of broken or lost appendages, but the nature of the effect differed among species (Fig. 2). In general, the number of broken appendages rose with the concentration of preservative ethanol. However, the magnitude of the effect varied dramatically, and only in some cases, it was significant. The lowest ethanol concentrations (30 or 50%) were also associated with an increased loss of appendages compared to intermediate concentrations. However, the difference between our "Gentle" and "Vigorous" treatments, with a two-fold difference in vortexing time, was relatively small.
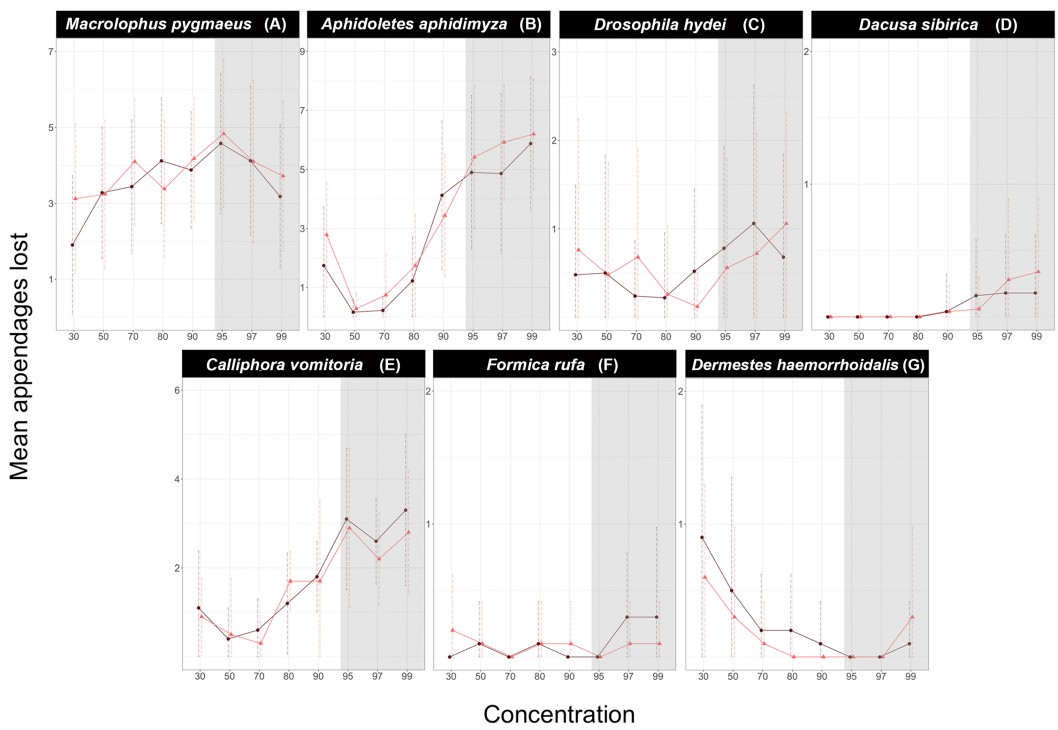

**Figure 2** **Effect of ethanol concentration on the number of appendages lost by each species.** (A) *Macrolophus pygmaeus*, (B) *Aphidoletes aphidimyza*, (C) *Drosophila hydei*, (D) *Dacusa sibrica*, (E) *Calliphora vomitoria*, (F) *Formica rufa*, (G) *Dermestes haemorrhoidalis*. Dark purple circles represent the Gentle shaking regime while bright red triangles represent the Vigorous regime. The shadowed area corresponds to the ethanol concentrations in which DNA is optimally preserved according to literature.

When fitted to the model, ethanol concentration alone had a very significant effect on the number of lost appendages in *Macrolophus*, *Aphidoletes,* and *Calliphora* but not in *Drosophila*, *Dacnusa* or *Dermestes* (*Formica* did not lose enough appendages to fit a model; Table 1, Table S2). In both *Aphidoletes* and *Calliphora*, intermediate concentrations of ethanol were optimal for the preservation of appendages, whereas the pattern in *Macrolophus* was less clear. In *Aphidoletes* and *Drosophila*, the shaking regime, as well as the interaction of concentration and shaking type, also had a significant effect on the number of appendages lost. Specifically, Vigorous shaking was more damaging at low and high concentrations of ethanol in both species. A similar trend, although not significant, could also be seen at high ethanol concentrations in *Dacnusa* and *Dermestes* (Fig. 1).

The remaining three species turned out to be very durable regardless of the ethanol concentration. In the case of *Formica*, only a few individuals lost one of the antennae. For *Dacnusa*, a small proportion lost antennae or, in one case, a wing, and only at ethanol concentrations >90%. *Dermestes* specimens sometimes lost elytra, particularly at low ethanol concentrations. However, for these three durable species, no significant differences were found based on the concentration, shaking regime, or their interaction.

**Table 1  Effect of concentration, treatment and their interaction in the number of lost appendages for each species in Experiment 1.** *F. rufa* did not produce enough data points to fit a model. Shown are Type III test of fixed effects if the model described.

| Effect | $\chi^2$ | Df | *p*-value |
|---|---|---|---|
| *Macrolophus pygmaeus* | | | |
| Concentration | 35.0614 | 7 | 0.0001*** |
| Treatment | 1.7642 | 1 | 0.1841 |
| Concentration:Treatment | 12.4535 | 7 | 0.0866 |
| *Aphidoletes aphidimyza* | | | |
| Concentration | 170.2500 | 7 | 0.0001*** |
| Treatment | 4.0480 | 1 | 0.0442* |
| Concentration:Treatment | 18.350 | 7 | 0.0104* |
| *Drosophila hydei* | | | |
| Concentration | 10.1877 | 7 | 0.1781 |
| Treatment | 5.3045 | 1 | 0.0212* |
| Concentration:Treatment | 15.7197 | 7 | 0.0278* |
| *Dacnusa sibirica* | | | |
| Concentration | 4.3189 | 7 | 0.7424 |
| Treatment | 0.0000 | 1 | 1.0000 |
| Concentration:Treatment | 3.9310 | 7 | 0.7877 |
| *Calliphora vomitoria* | | | |
| Concentration | 24.5232 | 7 | 0.0009*** |
| Treatment | 0.6392 | 1 | 0.4239 |
| Concentration:Treatment | 2.3005 | 7 | 0.9413 |
| *Dermestes haemorrhoidalis* | | | |
| Concentration | 11.5213 | 7 | 0.1174 |
| Treatment | 0.3203 | 1 | 0.5714 |
| Concentration:Treatment | 1.6935 | 7 | 0.9748 |

**Notes.**
Asterisks indicate levels of significance.
*$p$-value < 0.05.
**$p$-value < 0.01.
***$p$-value < 0.001.

## Experiment 2: Effects of transport regimes

In the transport experiment, only three species produced enough data points to fit a model: *Macrolophus*, *Aphidoletes,* and *Drosophila* (Fig. 3, Table 2, Table S3). For *Drosophila*, no significant differences were found between any transport treatment, ethanol concentration, or their interaction. This is consistent with the results from Experiment 1, showing that *Drosophila* is quite robust to the two different shaking regimes we tried under a broad range of ethanol concentrations, including the two concentrations (70% and 95%) we used in the transport experiment.

For the remaining two species, *Macrolophus* and *Aphidoletes*, specimens carried by a careful (Walking) technician sustained less damage than those carried by a reckless (Running) technician or sent by mail (PostNord). When the samples were treated with care (Walking), preservative ethanol concentration did not affect the number of appendages lost by either species. However, in those treated with less care, including Running and

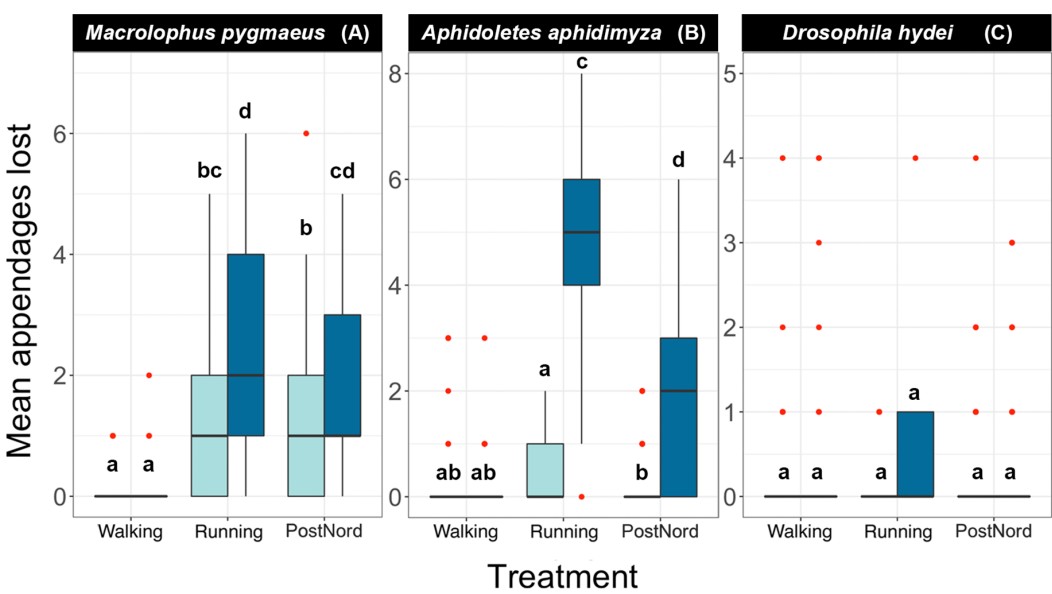

**Figure 3** **Effect of handling regimes and transport on the number of appendages lost by each species at two different concentrations of ethanol.** (A) *Macrolophus pygmaeus*, (B) *Aphidoletes aphidimyza*, (C) *Drosophila hydei*.Light blue boxes represent samples kept in 70% ethanol and dark blue samples kept in 95% ethanol. Red dots indicate outliers, and letters indicate groups of the Tukey Test of pairwise comparisons.

PostNord treatments, both species sustained significantly less damage when preserved in 70% ethanol.

## Experiment 3: Effects of freeze-thaw cycles and drying

As in the previous cases, the responses to freezing/drying treatments varied among species (Fig. 4, Table 3, Table S4). The three toughest species (*Dacnusa*, *Formica,* and *Dermestes*) had to be excluded from the analyses, as they did not generate enough data to fit a model. In the remaining species, we did not observe a significant treatment effect, that is, drying the specimens or exposing them to repeated cycles of freezing and thawing did not seem to affect their brittleness (Fig. 4). The only exception was *Macrolophus*, where we observed a mild *positive* effect of drying the specimens. Regardless of treatment, *Aphidoletes*, *Drosophila*, and *Calliphora* lost more appendages on average when stored at 95%, although the effect was only significant in *Aphidoletes*. In *Macrolophus*, there was no clear difference between the brittleness of specimens stored at 70% and at 95%. These species-specific responses to different ethanol concentrations closely match those seen in Experiment 1 (Fig. 2).

## Experiment 4: Effect of ethanol concentration on DNA preservation

For the 284 experimental libraries (278 insects and 6 blanks), we obtained a total of 6,061,251 read pairs. Of these, 5,444,2555 reads, for 276 insect libraries (range 948 –196,774, median 17,180 per sample), passed all analysis stages and were used for abundance comparisons (Table S7). Across libraries, the median abundance of the artificial target OTU relative to

**Table 2  Effect of concentration, treatment and their interaction in the number of lost appendages for each species in Experiment 2.** Only *M. pygmaeus*, *A. aphidimyza* and *D. hydei* produced enough data points to fit the model. Shown are Type III test of fixed effects if the model described.

| Effect | $\chi^2$ | Df | *p*-value |
|---|---|---|---|
| *Macrolophus pygmaeus* | | | |
| Concentration | 0.0000 | 1 | 1.0000 |
| Treatment | 19.3886 | 2 | 0.0006*** |
| Concentration:Treatment | 0.8273 | 2 | 0.6612 |
| *Aphidoletes aphidimyza* | | | |
| Concentration | 0.5238 | 1 | 0.4692 |
| Treatment | 11.5596 | 2 | 0.0031** |
| Concentration:Treatment | 21.5104 | 2 | 0.0002*** |
| *Drosophila hydei* | | | |
| Concentration | 0.9152 | 1 | 0.3387 |
| Treatment | 3.0386 | 2 | 0.2189 |
| Concentration:Treatment | 1.6271 | 2 | 0.4433 |

**Notes.**

Asterisks indicate levels of significance.

*$p$-value < 0.05.

**$p$-value < 0.01.

***$p$-value < 0.001.

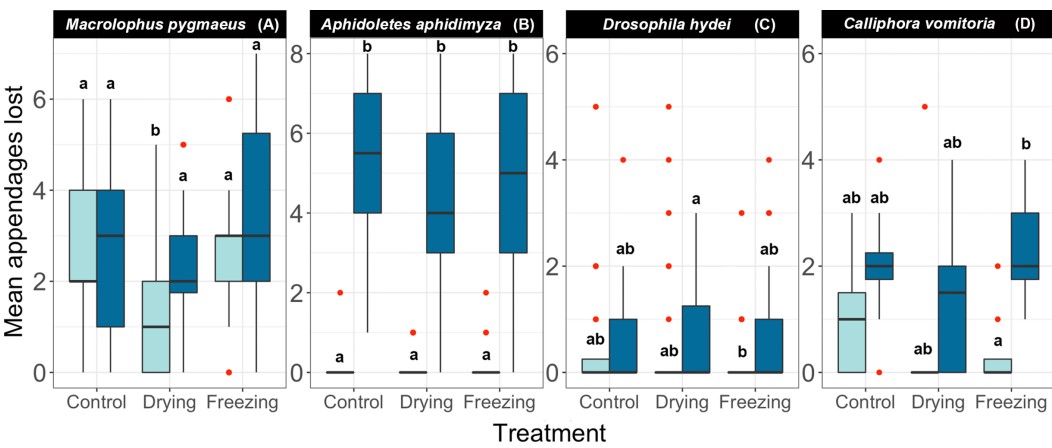

**Figure 4  Effect of different pre-treatments on the number of appendages lost by each species at two different concentrations of ethanol.** (A) *Macrolophus pygmaeus*, (B) *Aphidoletes aphidimyza*, (C) *Drosophila hydei*, (D) *Calliphora vomitoria*. All samples were shaken under a Gentle regime. Light blue boxes represent samples kept in 70% ethanol and dark blue boxes represent samples kept in 95% ethanol. Red dots indicate outliers, and letters indicate groups of the Tukey Test of pairwise comparisons.

the dominant OTU for the insect species was 0.136 (range 0.003–614); the value dropped slightly when all OTUs with >90% identity to the reference sequence were combined (0.134). Negative PCR and indexing controls had almost no COI reads (0–18), and in the

**Table 3** **Effect of concentration, treatment and their interaction in the number of lost appendages for each species in Experiment 3.** *D. haemorrhoidalis*, *F. rufa* and *D. sibirica* did not produced enough data points to fit the model. Shown are Type III test of fixed effects if the model described.

| Effect | $\chi^2$ | Df | *p*-value |
|---|---|---|---|
| *Macrolophus pygmaeus* | | | |
| Concentration | 0.1984 | 1 | 0.6560 |
| Treatment | 21.4882 | 2 | 0.0002[***] |
| Concentration:Treatment | 8.5936 | 2 | 0.0136[*] |
| *Aphidoletes aphidimyza* | | | |
| Concentration | 40.8062 | 1 | 0.0001[***] |
| Treatment | 1.1408 | 2 | 0.5653 |
| Concentration:Treatment | 1.5768 | 2 | 0.4546 |
| *Drosophila hydei* | | | |
| Concentration | 0.6375 | 1 | 0.4246 |
| Treatment | 6.0051 | 2 | 0.0496[*] |
| Concentration:Treatment | 6.3404 | 2 | 0.0419[*] |
| *Calliphora vomitoria* | | | |
| Concentration | 0.1544 | 1 | 0.6943 |
| Treatment | 4.5955 | 2 | 0.1005 |
| Concentration:Treatment | 6.1628 | 2 | 0.0459[*] |

**Notes.**
Asterisks indicate levels of significance.
[*]*p*-value < 0.05.
[**]*p*-value < 0.01.
[***]*p*-value < 0.001.

DNA extraction controls, most reads (1,760–8,610, >98.5% of the total) corresponded to the spike-in plasmid.

In all studied species, we found significant differences in the estimated COI target copy number between ethanol concentrations (Table S8) (Fig. 5). A gradual decrease in the number of COI copies with decreasing ethanol concentration is visible in all species, although the strength of the effect and the concentration at which it sets in varies (Fig. 5). Post-hoc analyses revealed a significant decrease in target copy numbers between at least some of the lower-concentration treatments and the 95% ethanol reference in all species. Specifically, the target copy number was significantly lower than in the 95% reference in samples preserved in 30% ethanol (seven out of seven species), 50% (seven species), 70% (five species), 80% (three species) and in one of the species preserved at 90% (*Macrolophus*). Samples preserved in 97% and 99% ethanol did not differ significantly from the 95% reference. The magnitude of the effect of the ethanol concentration varied among species: for example, in samples preserved at the lowest ethanol concentrations (30 and 50%), the decrease in the number of COI copies relative to the 95% reference ranged from 3 to 440-fold.

Results obtained from a dataset with all OTUs at >90% identity to reference sequence for that species considered do not differ qualitatively from the ones summarized above (Table S8).
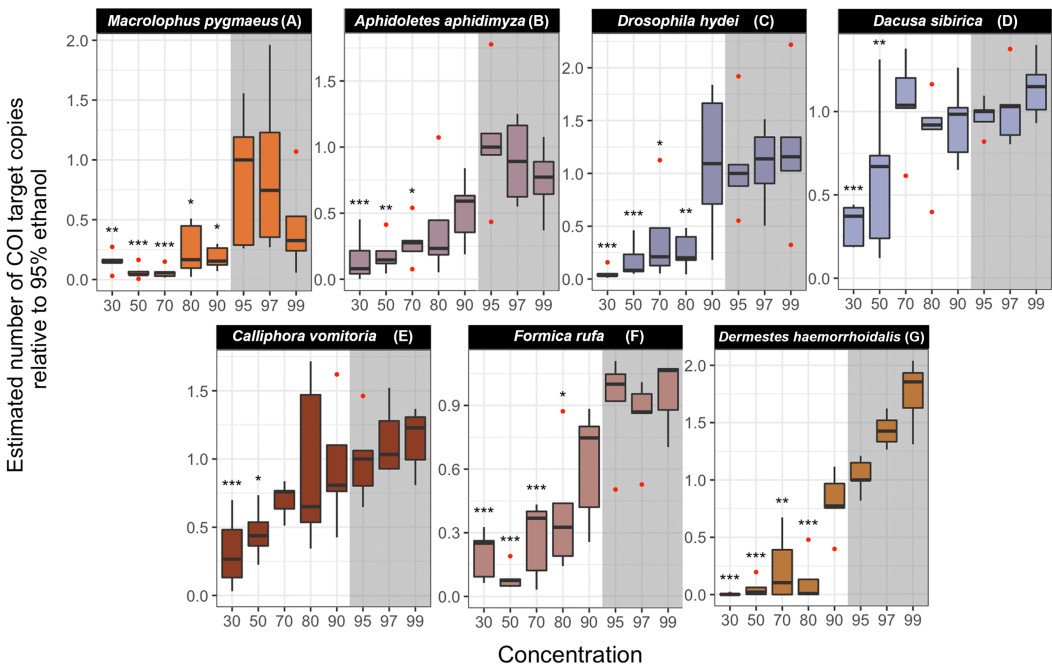

**Figure 5** **Estimated number of COI target copies in each species after one year of storage at room-temperature at eight ethanol concentrations.** (A) *Macrolophus pygmaeus*, (B) *Aphidoletes aphidimyza*, (C) *Drosophila hydei*, (D) *Dacusa sibrica*, (E) *Calliphora vomitoria*, (F) *Formica rufa*, (G) *Dermestes haemorrhoidalis*. The numbers were standardized relative to 95% ethanol concentration median. Barplots were based on five or four (in four cases) replicates. The concentrations in which the number of COI copies was statistically different from the 95% ethanol concentration treatment are indicated with asterisks (\*\*\* *p*-value < 0.001, \*\* *p* < 0.01, \* *p* < 0.05). The shadowed area corresponds to the ethanol concentrations in which DNA is optimally preserved according to literature.

## DISCUSSION

It is surprising that there are so few quantitative studies of the effect of ethanol concentration on the preservation of insects for morphological and molecular study. One possible reason for this is the difficulty of quantifying morphological preservation, but also the preservation of DNA for diverse sequencing-based applications. Our approaches allowed us to address both of these challenges.

The criterion we used here to evaluate morphological preservation, the number of lost appendages, has the major advantage that it is fast and easy to measure. However, it captures only one aspect of morphological preservation, namely brittleness. Brittleness is important in many contexts, for instance, when handling, examining or mounting specimens, but it is not an ideal measure of the preservation of fine morphological details, or the status of internal anatomy. For instance, we noted a clear discrepancy between brittleness and morphological preservation with the drying pre-treatment. Nevertheless, we believe our approach is a good starting point for further enquiry into the effect of ethanol concentration on the preservation of insects. Undoubtedly, it would be valuable to keep exploring more sophisticated measures of morphological conservation.

To assess DNA preservation, we used a quick measure of one simple aspect of DNA degradation: the absolute abundance of a certain 458-base long DNA fragment, part of the mitochondrial genome. It is clearly not a perfect measure of the overall DNA preservation. However, as DNA degradation happens primarily by the accumulation of random breaks within the DNA strands (although other processes play a role, for example, deamination of cytosines), it is unlikely that many longer fragments would remain once the abundance of 458-base targets decreases significantly. Thus, our results are of direct relevance to sequencing approaches that do not require high-molecular-weight DNA, but also informative to researchers that sequence long DNA fragments (e.g., PacBio or Oxford Nanopore sequencing).

The use of spike-in plasmids with artificial targets in amplicon sequencing experiments has been gaining popularity as a method for the quantification of microbiomes (*Tkacz, Hortala & Poole, 2018*; *Tourlousse et al., 2017*), while spike-in standards consisting of sequences of actual species have been used for abundance estimation in eukaryotic metagenomic studies (*Ji et al., 2020*). However, to our knowledge, they have not been used to assess DNA degradation. We found the absolute quantification of COI targets using spike-ins to be a quick, inexpensive, and, in our opinion, accurate and biologically informative way of gathering DNA preservation data.

In general, our results agree with expectations. They clearly show that ethanol concentration does have an impact on the fragility of insect specimens, even though the effects vary among taxonomic groups. High concentrations tend to increase brittleness, especially in weakly sclerotized insects. We also noted a tendency of more disruptive treatments to have a stronger effect on specimens preserved in high ethanol concentrations. Regarding DNA preservation, the expectations are also met, as we observed a lower proportion of suitable PCR-template DNA fragments in the DNA extracts of those individuals stored at lower ethanol concentrations after only one year.

The effect of ethanol concentrations on morphological preservation was most pronounced in the most weakly sclerotized species (*Aphidoletes* and *Calliphora* in particular). As the connective and muscular tissues are dried and made brittle by high ethanol concentrations, it is reasonable that insects with less exoskeletal anchorage in their joints will be more severely affected. However, interestingly, the negative effect of low ethanol concentrations was noticeable also in *Dermestes*. These beetles were very robust to damage at high ethanol concentrations, but tended to lose elytra –abdomen and head too –at low concentrations. The effect was not significant, but likely primarily because of lower numbers of *Dermestes* individuals (two per tube) compared to other species. Had we included more specimens, we suspect that this effect would have been significant. It would be interesting to test whether the increased fragility at lower ethanol concentrations is associated with the cuticle hardness that prevents preservative penetration into tissues, or perhaps decomposing microorganisms likely harbored by these carcass-feeding beetles (they were obtained from the dermestarium at the Swedish Museum of Natural History, where they clean off the soft tissue of vertebrate carcasses before the skeletal parts are included in the collections).

Our transport experiment clearly shows that the concentration of preservative ethanol has a strong effect on how fragile insect specimens are. Fragile specimens survive transport much better if stored in 70% ethanol than if stored in 95% ethanol (Fig. 3). The experiment also demonstrates that it is important to handle samples carefully. A careless technician can cause considerably more damage than shipping the specimens more than 400 km by a much-criticized national mail service. It should be noted, however, that all tubes were filled with liquid to nearly maximum capacity before being shipped, as generally recommended for safe transport (*Martin, 1977*). The results might have been different if the tubes had been half-empty.

Subjecting preserved specimens to repeated freezing and thawing cycles did not significantly affect their tendency to lose appendages. Similarly, drying the specimens and then reimmersing them in ethanol had no noticeable negative effect. In fact, drying had a slight positive effect on *Macrolophus* specimens, which lost fewer appendages after drying than in the control treatment. However, these results do not mean that the morphology is preserved intact through these treatments. For instance, we observed that many individuals from most species that had been dried had shrunken heads and abdomens that could complicate their taxonomic examination. It seems likely that also internal anatomy was affected both by the freezing-thawing cycles and by the drying-reimmersion treatment. Thus, it would be valuable to reinvestigate the effects of these treatments using other criteria for morphological preservation than the simple measure of appendage loss that we used.

As remarked previously, the effect of ethanol concentration varied strikingly among the different insect species we examined. Although a small set of species cannot be regarded as a comprehensive sample of insect diversity or fragility levels, some results may indicate patterns that apply more broadly to particular taxonomic groups of insects. For instance, it is notable that both hymenopterans we examined, *Dacnusa* and *Formica*, were quite robust regardless of treatment, despite the fact that *Dacnusa* is a small species of relatively delicate build. These results seem to be in concordance with previous studies that show that hymenopteran bodies resist the entrance of ethanol to the point that sometimes small holes must be carved in their exoskeleton to allow DNA preservation (*Dillon, Austin & Bartowsky, 1996*; *Mandrioli, 2008*). However, long-term storage in absolute ethanol can nevertheless make ants brittle (*King & Porter, 2004*). The three dipterans we examined (*Drosophila, Calliphora* and *Aphidoletes*) were all quite sensitive to the ethanol concentration, regardless of striking differences in size, body shape, and level of sclerotization. They were also consistently among the insects most affected by the different treatments we exposed them to. Coleoptera was represented by only one taxon in our study (*Dermestes*), but beetles are generally more sclerotized than other insects, and it is no surprise that *Dermestes* ranked among the most robust of the insects we studied. Hemiptera were similarly represented by a single taxon in our study (*Macrolophus*). This species was among the most fragile we studied, but the order does present a wide variety of body types, and future studies will have to show to what extent *Macrolophus* is representative. Our study did not include any representative of the order Lepidoptera, as these are rarely if ever preserved in ethanol when collected for morphological examination, and thus, the trade-off between morphological and molecular preservation in different concentrations of ethanol is of little relevance.

In general, the opposite pattern to the preservation of morphology was observed for the preservation of the DNA: the higher the ethanol concentration, the better preserved the DNA was. For most species, individuals preserved at concentrations of 80% ethanol or lower produced significantly fewer copies of the targeted PCR fragment than those preserved at 95%. This shows that, even when an amplification product can be obtained through PCR, there is, indeed, DNA degradation at these ethanol concentrations. Interestingly, concentrations of 97 and 99% ethanol never differ significantly from 95%, albeit in some species like *Dermestes* the average number of COI template copies was visibly higher at both of these concentrations. The concentration at which significant degradation of DNA was first observed varied among species. For four of the seven species, the preservation of the DNA was significantly worse from 70–80% and down, while for *Macrolophus,* even a concentration as high as 90% underperformed compared to 95%. On the other hand, the DNA of *Dacnusa* and *Calliphora* was not significantly degraded until the concentration was lowered to 50 or 30%. The mechanisms underlying these differences are unknown to us. One of the possible explanations offered above for the good morphological preservation of hymenopterans (namely, the resistance of their bodies to the entrance of ethanol) is directly at odds with the good preservation of DNA (extremely good in *Dacnusa*, quite good also in *Formica*). This suggests that the morphological solidity of hymenopterans is due to other causes. For instance, the body is more distinctly divided into hard sclerites in Hymenoptera than is the case for many other insect orders, like Diptera. Potentially, this could be associated with an decreased tendency to lose appendages.

Regardless of the causes, the variation among taxa in the effects of ethanol concentration on DNA preservation is highly relevant for metabarcoding studies. Metabarcoding involves the amplification of a barcoding gene sequence from DNA templates of many species simultaneously, so differences in the preservation of DNA among taxa could potentially result in significant biases, with the poorly preserved species contributing less than others to the pool of accessible template DNA copies. Minimally, this will cause problems with abundance estimation from metabarcoding data; in the worst case, poorly preserved species might go completely undetected, especially if they are rare.

Our study may be the first systematic study of the effect of ethanol concentration on morphological and molecular preservation of insects, albeit limited to only a handful of species. Our results largely confirm the commonly held belief that intermediate concentrations of ethanol, around 70–80%, are generally the best for morphological preservation, as high concentrations (above 90%) tend to make specimens fragile, whereas lower concentration in many cases allow some level of degradation. In contrast, the optimal DNA preservation is achieved at ethanol concentrations of >90%. Unfortunately this means that there is a conflict between preserving insects for morphological and for molecular work, as the ethanol concentrations that are ideal for the former purposes result in higher degradation rates of DNA. Is it possible to find a treatment that is ideal for both purposes? *Stein et al. (2013)* showed that if initial preservation is made in 95% ethanol, good PCR results can be obtained even after storing the insects in 70% ethanol for an extended period of time. However, this is the exact contrary of what is recommended for morphological study, where a gradual increase in the ethanol concentration is recommended to avoid fast

desiccation of the tissues. However, whether initial preservation in 95% and subsequent transfer to 70% for storage reduces the brittleness is unknown. The same could be said of the opposite procedure, also recommended by some experts: to maintain the specimens in $\geq$ 95 % ethanol for long-term storage, but to use lower concentrations (70 - 80 %) initially, or later for transport or manipulation; how these transferences impact DNA and morphological preservation should also be studied in the future. Recently, researchers have directed their attention towards preservatives based on propylene glycol (*Nakamura et al., 2020*; *Moreau et al., 2013*; *Robinson et al., 2020*). Lower flammability, evaporation, and cost make propylene glycol an attractive alternative to ethanol for trapping, transporting, and storing insect samples. Nevertheless, although these results look promising, more studies are needed to determine the quality of DNA and morphological preservation for different insect groups, under a range of conditions, for these preservatives.

For now, we do not have a preservation regimen that is ideal for both morphology and molecules. This must be taken into account when planning large collecting campaigns that aim to preserve material for both morphological and molecular work. For instance, we note that catches from Malaise traps, which are frequently used in insect inventories, contain a large portion of Diptera specimens (*Hebert et al., 2016*; *Karlsson et al., 2020*; *Ronquist et al., 2020*), a group whose morphology is often heavily affected by high ethanol concentrations. Hymenoptera, on the other hand, is abundant as well in Malaise traps, and not as prone to morphological damage. These differences need to be considered at the stage of experimental design: the relative importance of morphological against DNA data, or perhaps the focus on certain taxonomic groups, should guide the selection of preservative composition and preservation conditions. In any case, if Malaise trap inventory projects intend to preserve material for molecular and morphological study, it may be necessary to store the material at ethanol concentrations that are not ideal either for molecular or for morphological work.

## ACKNOWLEDGEMENTS

We are thankful to Dave Karlsson from Station Linné for receiving and returning the parcels with the tubes of the PostNord treatment, and to Peter Mortensen from the vertebrate collection at NRM for providing the beetle specimens. Ela Iwaszkiewisz contributed to the design of the artificial target for DNA preservation assessment, and Anna Michalik and Monika Prus prepared the plasmid for use.

### Funding

This project was supported by the European Union's Horizon 2020 research and innovation programme under the Marie Skłodowska-Curie grant agreement no. 642241 (BIG4 project), by the Knut and Alice Wallenberg Foundation (KAW 2017.088), Polish National Agency for Academic Exchange grant PPN/PPO/2018/1/00015 and Polish National Science Centre grant 2018/31/B/NZ8/01158. The funders had no role in study design, data collection and analysis, decision to publish, or preparation of the manuscript.

## Grant Disclosures

The following grant information was disclosed by the authors:

European Union's Horizon 2020 research and innovation programme: 642241.

Knut and Alice Wallenberg Foundation: KAW 2017.088.

Polish National Agency for Academic Exchange: PPN/PPO/2018/1/00015.

Polish National Science Centre: 2018/31/B/NZ8/01158.

## Competing Interests

The authors declare there are no competing interests.

## Author Contributions

- Daniel Marquina conceived and designed the experiments, performed the experiments, analyzed the data, prepared figures and/or tables, authored or reviewed drafts of the paper, and approved the final draft.
- Mateusz Buczek performed the experiments, analyzed the data, authored or reviewed drafts of the paper, and approved the final draft.
- Fredrik Ronquist conceived and designed the experiments, authored or reviewed drafts of the paper, and approved the final draft.
- Piotr Łukasik conceived and designed the experiments, performed the experiments, analyzed the data, authored or reviewed drafts of the paper, and approved the final draft.

## DNA Deposition

The following information was supplied regarding the deposition of DNA sequences:

The amplicon sequencing data are available in NCBI Short Read Archive: PRJNA662328.

## Data Availability

All raw appendages counts, sequence reads counts and insert provenance are available in the Supplementary Files.

## Supplemental Information

Supplemental information for this article can be found online at http://dx.doi.org/10.7717/peerj.10799#supplemental-information.

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
