# Peer review of "The effect of ethanol concentration on the morphological and molecular preservation of insects for biodiversity studies"

_PeerJ, doi:10.7717/peerj.10799_

## Round 0.1 · original submission · Minor Revisions

The manuscript reports a comprehensive study of the effect of ethanol concentration on morphological and molecular preservation of insects. Seven species from four insect orders with different level of sclerotization are sampled in this study. Although this sampling set seems not large, the authors designed systematic and impressive experiments to quantitatively test the effect of ethanol concentration. I think many colleagues will be interested in this general topic and their results.

Both reviewers recommended Minor revisions. They also provided concerns and suggestions on the sampling and discussion of the manuscript. For example, one reviewer suggested the authors should pay attention to the description about limitations of your sampling and related conclusion. Another reviewer provided helpful recommendation on ethanol concentrations. I think the authors can address these and other comments in the revision.

Reviewer 1 ·

Basic reporting

no comment

Experimental design

Seven species were investigated by the authors. I noticed that some of them are the ones whose individual integrity are more sensitive to the concentration of ethanol in preservation. That means the results and conclusion are highly affected by which representative species are studied.

Validity of the findings

With just seven species representing four orders, I don't think a proper general conclusion for insects can be made. Therefore, I recommand the authors to limit the conclusion of the positive results to "at least n percent of insect species". For example, Miridae is the largest family in Heteroptera and its species account for about 1/4 of all true bugs species. Besides, no moth or butterfly species was included in this study. No taxonomist will use the alcohol kept specimens for morphological studies of Lepidoptera, no matter what concentration of alcohol is. Therefore, the finding of "trade-off" is partially right, and the authors are responsible to make the readers be aware of such situation.

Reviewer 2 ·

Basic reporting

The manuscript is very well written and clear throughout. The context is explained well and is properly documented with respect to the literature. It seems like a long manuscript for what is otherwise a mostly straight-forward report: it could probably be shortened, although I would leave it as is if there is any risk of losing readability.

It is easier to amplify extracted DNA from dried specimens than from ethanol-preserved specimens, as has been documented frequently in the many studies trying to sequence DNA from older museum specimens. There are other, more important reasons specimens for DNA study are often stored in ethanol, but then there is also the mistaken belief that ethanol-preservation is best. The sentence on lines 58-61 is over-stated and needs to be revised in order not to promote this false notion.

Experimental design

The research questions are clearly presented and tested. The four experiments are clearly presented and the application of spike-in standards for amplicon quantification method in is particularly interesting.

Validity of the findings

The findings quantifiably support the commonly held belief that specimen and DNA integrity are inversely affected by ethanol preservative concentration. The findings are all reasonably supported.

The conclusions of the paper suggest trying to find a compromise given the needs of the researcher and the taxonomy of the insect specimens. However, the authors neglect another possibility, that is, of transferring material between ethanol concentrations for different purposes. For example, my standard practice is to collect field material into 70% for transport, transfer it to 95% for long term storage in the lab, but then to re-transfer it back to 70% whenever I need to manipulate or transport the specimen. This increases handling time, of course, but it also seeks to have the best of both worlds regarding specimen and DNA integrity.

---

## Round 0.2 · accepted · Accept

I think the issues raised by reviewers have been addressed. The manuscript can be accepted for publication now. Interesting paper. Congratulations!